# ON THE INTERPLAY BETWEEN LEARNING AND MEMORY IN DEEP STATE SPACE MODELS

## ABSTRACT

Deep state-space models (SSMs) have emerged as a powerful deep learning architecture for sequence modeling, but the theory of how these models learn long-term dependencies lags the practice. To explain how parameterization and the number of layers affect a model's expressiveness, we study the properties of deep *linear* SSMs, i.e., linearly coupled stacks of linear time-invariant systems. We show that such systems share timescales across layers, and we provide novel analysis on the role of linear feedforward connections in regularizing these temporal dependencies. In practice, SSMs can struggle with an explosion of the hidden state variance when learning long-term dependencies. We expand our theoretical understanding of this problem for deep SSMs and provide new intuitions on how this problem may be resolved by increasing the number of layers. Finally, we confirm our theoretical results in a teacher-student framework and show the effects of model parameterization on learning convergence.

## 1 INTRODUCTION

Deep state space models (SSMs) have achieved impressive performance on a variety of long-range sequence modeling tasks. They are competitive with state-of-the-art models for natural language (Fu et al., 2023; Gu & Dao, 2023), image tasks (Nguyen et al., 2022; Zhu et al., 2024), audio processing (Goel et al., 2022), video forecasting (Smith et al., 2024), reinforcement learning (Lu et al., 2024), genetic sequence prediction (Schiff et al., 2024), and other time-series data (Zhou et al., 2023; Patro & Agneeswaran, 2024). At their core, deep SSMs are remarkably simple — each layer applies a linear filter to its inputs, then the outputs are nonlinearly transformed and passed to the next layer. The simplicity of the architecture suggests these models may be amenable to theoretical treatment, and recent work has taken steps in this direction (Wang et al., 2023; Orvieto et al., 2024; Cirone et al., 2024). However, the mechanisms explaining how deep SSMs learn long-range dependencies is still not well understood.

One key question is how the number of layers (*depth*) and latent state size (*width*) affect a model's ability to learn long-range dependencies in sequential data. Learning such dependencies is challenging due to issues of vanishing and exploding gradients (Bengio et al., 1994) and increased sensitivity to parameter changes as the model encodes long timescale dependencies (Zucchet & Orvieto, 2024).

We address this question by considering the simplified setting of deep *linear* SSMs. Inspired by prior theoretical analyses of deep linear feedforward networks (Baldi & Hornik, 1989; Saxe et al., 2014; 2019), we consider deep SSMs where the nonlinearities between layers are removed. Recent work has studied the learning dynamics of single layer linear SSMs and shown that the analytical solutions derived from such models can provide intuition for their nonlinear counterparts (Zucchet & Orvieto, 2024; Smékal et al., 2024). While a deep linear SSM can also be cast as a single layer model, we show that the learning dynamics of deep linear models vary with memory expressivity.

Specifically, we study how the memory of a deep linear SSM, as measured by properties of its impulse response, autocorrelation, and transfer functions, vary with the depth and width of the model. These properties lead to predictions of how learning dynamics will vary across architectures, which we test empirically in a teacher-student setting (Hardt et al., 2019; Zucchet & Orvieto, 2024). We find regimes in which increasing depth may lead to faster convergence than increasing width, and discuss practical extensions of our theory to deep SSMs with nonlinear transformations between layers.

## 2 BACKGROUND AND RELATED WORK

Deep SSMs consist of stacks of state space layers with nonlinear coupling in between. Each state space layer maps an input sequence $u_{1:T}$ to an output sequence $y_{1:T}$ via a linear filter,

$$x_t = A_t x_{t-1} + B_t u_t, \qquad y_t = C_t x_t, \qquad (1)$$

where $x_t \in \mathbb{R}^N$ are the latent states. The outputs $y_t$ are passed through a nonlinearity before becoming the inputs to the next layer. We consider the special case of linear time-invariant (LTI) systems in which $A_t \equiv A$, $B_t \equiv B$, and $C_t \equiv C$, as in several deep SSMs (Gu et al., 2022a; Smith et al., 2023). Furthermore, we focus on single-input ($u_t \in \mathbb{R}$), single-output ($y_t \in \mathbb{R}$) SSMs parameterized by a real-valued, diagonal state-transition matrix $A \in \mathbb{R}^{N \times N}$, input vector $B \in \mathbb{R}^{N \times 1}$, and output vector $C \in \mathbb{R}^{1 \times N}$, as in S4D (Gu et al., 2022b).

Following the empirical success of deep SSMs, many questions have emerged about their temporal expressivity and how they compare to other sequence modeling architectures. A growing body of theoretical work has elucidated their expressive power and potential limitations. Orvieto et al. (2024) and Wang & Xue (2023) have shown that state space layers with nonlinear coupling in between are sufficiently expressive to approximate any sequence-to-sequence map, and Cirone et al. (2024) provided an extended analysis on the expressivity benefits of input-controlled state space dynamics.

Recent work has made theoretical progress toward analyzing the learning dynamics of single-layer SSMs. In this case, the problem reduces to learning the parameters of a linear dynamical system with gradient descent, as studied by Hardt et al. (2019). While the linear recurrences of deep SSMs make it easier to control the issue of vanishing and exploding gradients (Bengio et al., 1994), Zucchet & Orvieto (2024) showed that even outside the exploding regime, the latent states $x_t$ become increasingly sensitive to parameter changes as the magnitude of the eigenvalues $\lambda$ of the dynamics matrix $A$ approach one. They refer to this problem as the *curse of memory*. It leads to sharp loss landscapes which make gradient-based optimization particularly challenging. In practice, this challenge has been addressed at the layer level with careful discretization techniques, like those used in the S4 and S5 architectures (Gu et al., 2022a; Smith et al., 2023), or explicit input regularization such as that used in the LRU (Orvieto et al., 2023).

However, the curse of memory has not been studied for deep SSMs. The nonlinearities between layers render such models analytically intractable. Here, we follow previous work on analyzing deep linear feedforward networks (Baldi & Hornik, 1989; Saxe et al., 2014; 2019) and study deep *linear* SSMs. With the nonlinearity between layers removed, we can characterize the effective memory of these models and its effects on learning dynamics. We show that these considerations have practical impact on parameterization choices of depth and width in deep SSMs.

## 3 THEORETICAL RESULTS

We consider three ways of formalizing the memory of a deep linear SSM: in terms of the autocorrelation of successive layers' outputs; in terms of the group delay, which is derived from the frequency response function; and in terms of the eigendecomposition of an equivalent 1-layer linear SSM. We use these formalisms to study how depth and width affect learning and memory in deep SSMs.

### 3.1 MEMORY AND THE AUTOCORRELATION FUNCTION

One way to characterize the memory of a state space layer is in terms of the autocorrelation function (ACF) of its outputs. The ACF has historically been applied to time-series data to identify an appropriate statistical model for forecasting (Shumway & Stoffer, 2017). Here, we are take a different approach and apply the ACF to interpret the *outputs* of a given model. Intuitively, the ACF depends on two factors: the autocorrelation of the layer's inputs and the eigenvalues of its dynamics matrix. When the eigenvalues are close to one, the layer will exhibit long-timescale dynamics, and its outputs will have strong autocorrelations. When the inputs are autocorrelated, we expect the outputs to be as well. Moreover, as Zucchet & Orvieto (2024) showed, the variance of a layer's latent states and gradients scales with the autocorrelation of the inputs. Thus, understanding how autocorrelation varies across layers of a deep SSM is paramount to understanding its memory and learning dynamics.

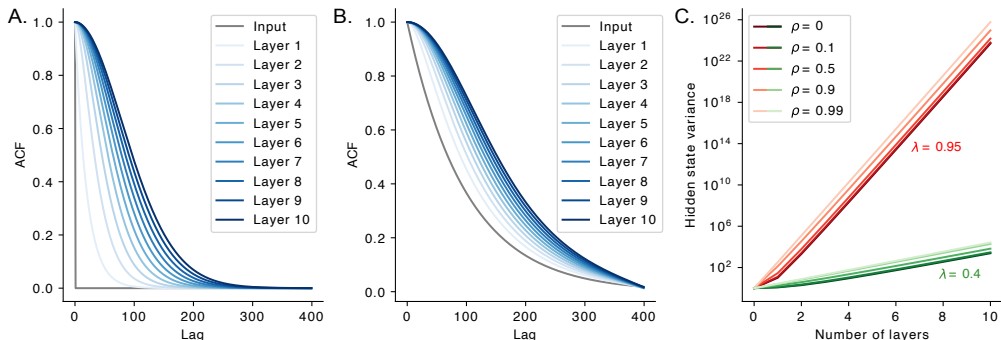

Figure 1: Inputs across layers of a deep linear SSM become more correlated, leading to an explosion of latent state variance called the *curse of memory*. The plot shows the evolution of the normalized autocorrelation function and hidden state variance across layers for different initial inputs, $\rho = 0$ corresponds to uncorrelated inputs, $\rho = 1$ corresponds to fully correlated inputs. **A.** Normalized auto-correlation function (ACF) of white-noise inputs $u_t$ and of the outputs of a stack of linear state space layers with $\lambda = 0.9$ at each layer. **B.** ACF across the outputs of 10 layers with the initial inputs following $R_u(\Delta) = \rho^{|\Delta|}$, with $\rho = 0.95$. **C.** Latent state variance grows exponentially with the number of layers in a deep linear SSM for different correlations of the input and two fixed timescales of the layers in the model ($\lambda = 0.95$ and $\lambda = 0.4$). Long-term dependencies corresponding to larger timescales result in greater explosion of latent state variance across layers. Hidden state variance eventually evolves as if $\rho = 1$ regardless of the initial ACF of the inputs, shedding light on the limitations of the analysis obtained from considering layer-wise latent state variance without accounting for how the ACF changes across layers.

Let $R^{(k)}(\Delta)$ denote the autocorrelation function of the outputs of layer $k$, or equivalently as the autocorrelation function of the inputs to layer $k + 1$, with $R_x(\Delta) = \mathbb{E}[x_{t+\Delta} x_t]$. We generally omit the subscript $x$. As in previous work (Zucchet & Orvieto, 2024), we consider the $N = 1$ dimensional case of scalar latent states with dynamics $A^{(k)} = \lambda^{(k)}$ at layer $k$. Furthermore, we assume $B^{(k)} = C^{(k)} = 1$ (we will consider the importance of these parameters in section 3.3). Proposition 1 expresses $R^{(k)}(\Delta)$ as a function of $R^{(k-1)}(\Delta)$.

**Proposition 1.** *Assuming the inputs to layer $k$ are wide-sense stationary with auto-correlation function $R^{(k-1)}(\Delta)$, the auto-correlation function of the layer's output is,*

$$R^{(k)}(\Delta) = \frac{1}{1 - (\lambda^{(k)})^2} \left( R^{(k-1)}(\Delta) + \sum_{\Delta' \geq 1} (\lambda^{(k)})^{\Delta'} \left( R^{(k-1)}(\Delta + \Delta') + R^{(k-1)}(\Delta - \Delta') \right) \right).$$

$$(2)$$

The proof of proposition 1 is in Appendix A. Applying this proposition recursively with the initial condition $R^{(0)}(\Delta) = R_u(\Delta)$ yields the ACF of the inputs to each layer of a deep SSM.

Figure 1A-B shows the evolution of the autocorrelation function across a stack of 10 linear recurrent layers for two different initial inputs, with autocorrelation function given by $R_u(\Delta) = \rho^{|\Delta|}$. Figure 1C shows that regardless of the initial autocorrelation of the inputs, which is determined by $\rho$, the outputs of the network become more correlated with each successive layer. In this example, after only a few layers, the hidden state variance grows at the same exponential rate for all $\rho$.

Note that $R^{(k)}(\Delta)$ contains a factor $\prod_{j=1}^{k} \left[ \frac{1}{1-(\lambda^{(j)})^2} \right]$. The ACF, and hence the latent state variance of layer $k$, depend not only on the parameters of that layer but of all its preceding layers as well. When all layers encodes long-term dependencies corresponding to $\lambda^{(k)} \to 1$, the hidden state variance grows exponentially with depth. This theoretical observation suggests that if any layer learns a long term dependency by setting $\lambda^{(k)} \approx 1$, it will lead the latent state and gradient variance to diverge. To compensate, the deep SSM may set other $\lambda^{(j)}$ for $j \neq k$ to close to zero in order to control the latent state variance and mitigate the exponential growth in fig. 1C. We test this prediction in Section 4.

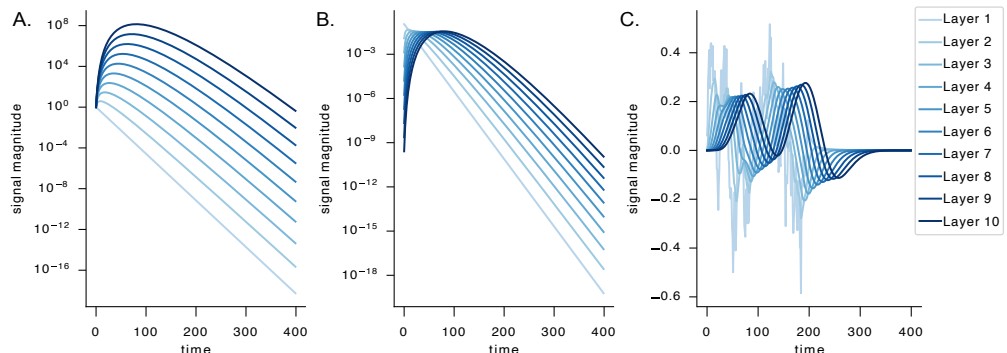

Figure 2: Adding layers to a stack of linear SSMs increases the effective memory of the system and the correlation of outputs, suggesting existing theory describing the memory of deep SSMs and the evolution of hidden state variance is incomplete. The plot shows signal propagation across a scalar deep linear SSM with $\lambda = 0.9$ at each layer. **A.** Adding successive LTI subsystems amplifies the original impulse signal and elongates the output decay. **B.** Same as **A** but with input normalization between layers, showing the maximum impulse response shifts forward in time with the addition of more layers. **C.** Effect of cascaded LTI systems on the propagation of white-noise input. Each successive layer introduces a time-delay and acts as a smoothing filter.

### 3.2 MEMORY AND THE GROUP DELAY

The autocorrelation function offers a useful measure of the memory of a state space layer, but it conflates the effect of the latent state dynamics with the autocorrelation of the inputs. Another way to characterize the memory of a linear system is in terms of the *group delay*, which intuitively captures the time it takes an input to percolate to the output of a system. The group delay is defined as a function of the frequency response of the system, which is given below for deep linear SSMs.

**Proposition 2.** *Following Smékal et al. (2024), let $U_l \in \mathbb{C}$ and $Y_l \in \mathbb{C}$ for $l = 1, \ldots, L$ denote the discrete Fourier transform (DFT) of the inputs $u_{1:T}$ and outputs $y_{1:T}$, respectively. For diagonal dynamics matrices $A = \mathrm{diag}(\lambda_1, \ldots, \lambda_N)$ with $|\lambda_n| < 1$ for all $n = 1, \ldots, N$ to ensure stability, the SSM outputs from equation 1 are transformed from a recurrence in the time domain to a multiplication by the system's frequency response in the frequency domain, $Y_l = H_l U_l$, where $H_l \in \mathbb{C}$ is given by,*

$$H_l = C G_l B, \qquad\qquad G_l = (I - A e^{-j\omega_l})^{-1}. \tag{3}$$

*The frequency response of a deep linear SSM is the product of frequency responses for each layer,*

$$H_l = \prod_{k=1}^{K} C^{(k)} G_l^{(k)} B^{(k)}. \tag{4}$$

The proof of Proposition 2 is in Appendix B.

The group delay, also known as the average time delay or mean delay in continuous-time LTI systems, represents the average time it takes for a change in the input to propagate through all the layers and affect the output. Alternatively, it can be interpreted as the time during which an input can produce a change in latent state and affect the output. Both perspectives suggest that group delay is a reasonable proxy for a system's memory.

Technically, the group delay, $\tau_d$, is defined as the negative of the derivative of the phase of the transfer function with respect to frequency, evaluated at zero. For simplicity, consider the 1-dimensional case with $C^{(k)} = B^{(k)} = 1$ for all $k$, and let $A^{(k)} = \lambda^{(k)}$. Under these assumptions, the group delay is,

$$\tau_d = \sum_{k=1}^{K} \frac{\lambda^{(k)}}{1 - \lambda^{(k)}}. \tag{5}$$

We see that the group delay scales linearly with the delay of each layer. The full derivation of the group delay can be found in Appendix B.

Figure 2A and B show the increasing delay across layers for an impulse signal propagating through a stack of ten linear 1-dimensional layers, each with a fixed eigenvalue $\lambda = 0.9$. The delay corresponds to the signal moving to the right along the time axis. Figure 2C shows this effect for white noise input. In practice, this means that by adding more layers to a deep SSM, the model should be able to capture longer-term dependencies.

### 3.3 MEMORY AND THE EIGENDECOMPOSITION OF AN EQUIVALENT ONE-LAYER MODEL

Ultimately, the eigenvalues of the dynamics matrices determine the memory of the system, as measured by both the autocorrelation function and the group delay. For a deep linear SSM, the entire stack of layers can be cast as a single, linear state space layer (Oppenheim, 1999). A third way to characterize the memory of a deep linear SSM is in terms of the eigenvalues of the dynamics matrix of the equivalent one-layer system, as described below.

Consider a deep linear SSM with $K$ layers and $N = 1$ dimensional states with $B^{(k)} = C^{(k)} = 1$ for all $k$,

$$
\begin{aligned}
x_{t+1}^{(1)} &= \lambda^{(1)} x_t^{(1)} + u_{t+1} & y_t^{(1)} &= x_t^{(1)} \\
x_{t+1}^{(2)} &= \lambda^{(2)} x_t^{(2)} + y_{t+1}^{(1)} & y_t^{(2)} &= x_t^{(2)} \\
&\;\;\vdots & &\;\;\vdots \\
x_{t+1}^{(K)} &= \lambda^{(K)} x_t^{(K)} + y_{t+1}^{(K-1)} & y_t^{(K)} &= x_t^{(K)}
\end{aligned}
\tag{6}
$$

This model can be expressed as a single-layer model with $K$-dimensional states,

$$
\mathbf{x}_{t+1} = \mathbf{A}\mathbf{x}_t + \mathbf{B}u_{t+1}, \qquad\qquad y_t^{(K)} = \mathbf{C}\mathbf{x}_t,
\tag{7}
$$

$$
\mathbf{A} = \begin{bmatrix}
\lambda^{(1)} & 0 & 0 & \cdots & 0 \\
\lambda^{(1)} & \lambda^{(2)} & 0 & \cdots & 0 \\
\lambda^{(1)} & \lambda^{(2)} & \lambda^{(3)} & \cdots & 0 \\
\vdots & \vdots & \vdots & \ddots & \vdots \\
\lambda^{(1)} & \lambda^{(2)} & \lambda^{(3)} & \cdots & \lambda^{(K)}
\end{bmatrix}, \qquad
\mathbf{B} = \begin{bmatrix} 1 \\ 1 \\ 1 \\ \vdots \\ 1 \end{bmatrix}, \qquad
\mathbf{C} = \begin{bmatrix} 0 & 0 & 0 & \cdots & 1 \end{bmatrix},
\tag{8}
$$

where we use the bold notation $\{\mathbf{A}, \mathbf{B}, \mathbf{C}, \mathbf{x}_t\}$ to explicitly distinguish this representation from the parameters of preceding sections.

Importantly, the resulting system matrix is lower triangular, not diagonal. However, when the $\{\lambda^{(k)}\}$ are distinct, the dynamics matrix can be diagonalized as $\mathbf{A} = \mathbf{P}\mathbf{\Lambda}\mathbf{P}^{-1}$, where $\mathbf{P}$ is the matrix of eigenvectors of $\mathbf{A}$ and $\mathbf{\Lambda} = \mathrm{diag}(\lambda^{(1)}, \ldots, \lambda^{(K)})$ is the diagonal matrix of eigenvalues of $\mathbf{A}$. We diagonalize the linear system in eq. (7) by introducing a new latent variable $\mathbf{z}_t = \mathbf{P}^{-1}\mathbf{x}_t$ so that,

$$
\mathbf{z}_{t+1} = \mathbf{\Lambda}\mathbf{z}_t + \widetilde{\mathbf{B}}u_{t+1}, \qquad y_t^{(K)} = \widetilde{\mathbf{C}}\mathbf{z}_t.
\tag{9}
$$

where $\widetilde{\mathbf{B}} = \mathbf{P}^{-1}\mathbf{B}$ and $\widetilde{\mathbf{C}} = \mathbf{C}\mathbf{P}$.

Previous sections argued that the memory of deep linear SSMs, as measured by the ACF and group delay, grows with depth. Here, we constructed an equivalent single-layer model, but we found that the eigenvalues of the diagonalized system eq. (9) are exactly the same as those of the constituent layers. How can the system exhibit longer memory if the eigenvalues do not become closer to one? The answer is that the expressions for the ACF and group delay hid an implicit dependence on $\widetilde{\mathbf{B}}$ and $\widetilde{\mathbf{C}}$ by considering the special case where both are one. In this diagonalized, single-layer system, however, we have to consider how $\widetilde{\mathbf{B}}$ and $\widetilde{\mathbf{C}}$ are influenced by the eigenvectors $\mathbf{P}$.

Examining $\mathbf{P}$ and $\mathbf{P}^{-1}$ more closely, we see that they are highly structured, thanks to the lower-triangular structure of $\mathbf{A}$. In the $K = 4$ case, for example,

$$
\mathbf{P} = \begin{pmatrix}
\frac{(\lambda_1-\lambda_2)\lambda_3(\lambda_4-\lambda_1)}{\lambda_1^3} - \frac{\lambda_1^2-\lambda_1\lambda_2-\lambda_1\lambda_4+\lambda_2\lambda_4}{\lambda_1^2} & 0 & 0 & 0 \\
\frac{-(\lambda_1^2-\lambda_1\lambda_3-\lambda_1\lambda_4+\lambda_3\lambda_4)}{\lambda_1^2} & \frac{-(\lambda_2^2-\lambda_2\lambda_3-\lambda_2\lambda_4+\lambda_3\lambda_4)}{\lambda_2^2} & 0 & 0 \\
\frac{-(\lambda_4-\lambda_1)}{\lambda_1} & \frac{-(\lambda_4-\lambda_2)}{\lambda_2} & \frac{-(\lambda_4-\lambda_3)}{\lambda_3} & 0 \\
1 & 1 & 1 & 1
\end{pmatrix}
\tag{10}
$$

$$
\mathbf{P}^{-1} = \begin{pmatrix}
\frac{\lambda_1^3}{(\lambda_1-\lambda_2)(\lambda_1-\lambda_3)(\lambda_1-\lambda_4)} & 0 & 0 & 0 \\
\frac{-\lambda_1\lambda_2^2}{(\lambda_1-\lambda_2)(\lambda_2-\lambda_3)(\lambda_2-\lambda_4)} & \frac{-\lambda_2^2}{(\lambda_2-\lambda_3)(\lambda_2-\lambda_4)} & 0 & 0 \\
\frac{-\lambda_1\lambda_3^2}{(\lambda_1-\lambda_3)(\lambda_3-\lambda_2)(\lambda_3-\lambda_4)} & \frac{\lambda_2\lambda_3}{(\lambda_2-\lambda_3)(\lambda_3-\lambda_4)} & \frac{\lambda_3}{\lambda_3-\lambda_4} & 0 \\
\frac{-\lambda_1\lambda_4^2}{(\lambda_1-\lambda_4)(\lambda_4-\lambda_2)(\lambda_4-\lambda_3)} & \frac{\lambda_2\lambda_4}{(\lambda_2-\lambda_4)(\lambda_4-\lambda_3)} & \frac{-\lambda_3}{\lambda_3-\lambda_4} & 1
\end{pmatrix}.
\tag{11}
$$

The form of $\mathbf{P}^{-1}$ shows that the inputs are projected onto all of the eigenmodes of the diagonalized system in proportion to the eigenvalues of the system and inversely proportional to the gaps between them. Likewise, the outputs are read out from all modes equally, since $\widetilde{\mathbf{C}} = \mathbf{1}_K$. The projections onto and out from these eigenmodes, together with the associated eigenvalues, determine the memory of the deep linear SSM.

## 4 EMPIRICAL RESULTS IN THE TEACHER-STUDENT SETTING

We used simple simulations to test the effects of depth and width on the memory of deep SSMs. To have full control over the types of long-term dependencies, we train our deep SSMs in a teacher-student setting (Hardt et al., 2019; Zucchet & Orvieto, 2024). This approach also allows us to compare the results of learning when the student and teacher differ in number of layers, latent state size, ablations on which components of the model are learnable, etc. The inputs to each model are sampled from uncorrelated white noise, $u_t \sim \mathcal{N}(0, 1)$. The student is trained with the Adam optimizer on the mean-squared error loss between the teacher's outputs and its outputs.

### 4.1 MEMORY OF DEEP SSMS AND THE ROLE OF $B$ AND $C$

In the analysis of Section 3.1, we hypothesized that long-term dependencies would likely be concentrated to only one layer in a deep SSM to prevent the explosion of latent and gradient variances. In Section 3.2, we found that stacking multiple linear state space models increases the group delay of the full system. To study these effects in the teacher-student task, we first consider a setting where only the dynamics matrices $A^{(k)}$ are trainable across layers and $B$ and $C$ are fixed, as in the corresponding theoretical results.

We trained a 2-layer, 2-dimensional student to match the outputs of a 1-layer, 2-dimensional teacher. In this experiment, we chose a 1-layer teacher to precisely specify the timescale of the task given by the eigenvalues of the teacher $\lambda_* = \{0.9, 0.99\}$ (for a multi-layer teacher, we would need to resort to calculating its group delay to fully understand the nature of the long-term dependency). The left-most panel in fig. 3A shows the learning trajectories of the eigenvalues of the diagonal $A$ matrices of the student, the trajectories for $B$ and $C$ are constant because they were fixed. The small horizontal bars in all subplots show the parameters of the teacher. We can therefore compare the learned parameters of the student at each layer with those of the 1-layer teacher. In this fixed $B$, $C$ experiment, the learned eigenvalues across the two layers of the student are distributed as our theoretical results would predict for this simplified case. The first layer, $A^{(0)}$, learned two non-zero eigenvalues with either one smaller than the corresponding teacher eigenvalue ($\{\lambda_1^{(0)} = 0.98, \lambda_2^{(0)} = 0.61\}$). The second layer, as expected given the first layer, learned comparably small eigenvalues ($\{\lambda_1^{(1)} = -0.33, \lambda_2^{(1)} = -0.33\}$), consistent with our prediction. Taken together as a single 2-layer system, the student has converged to a stable set of parameters approximating the teacher's outputs.

Next, we analyzed the role of the feed-forward projections $B$ and $C$, and posited that they play a role in the memory of deep SSMs. We confirm this effect in the same experiment from fig. 3A, now

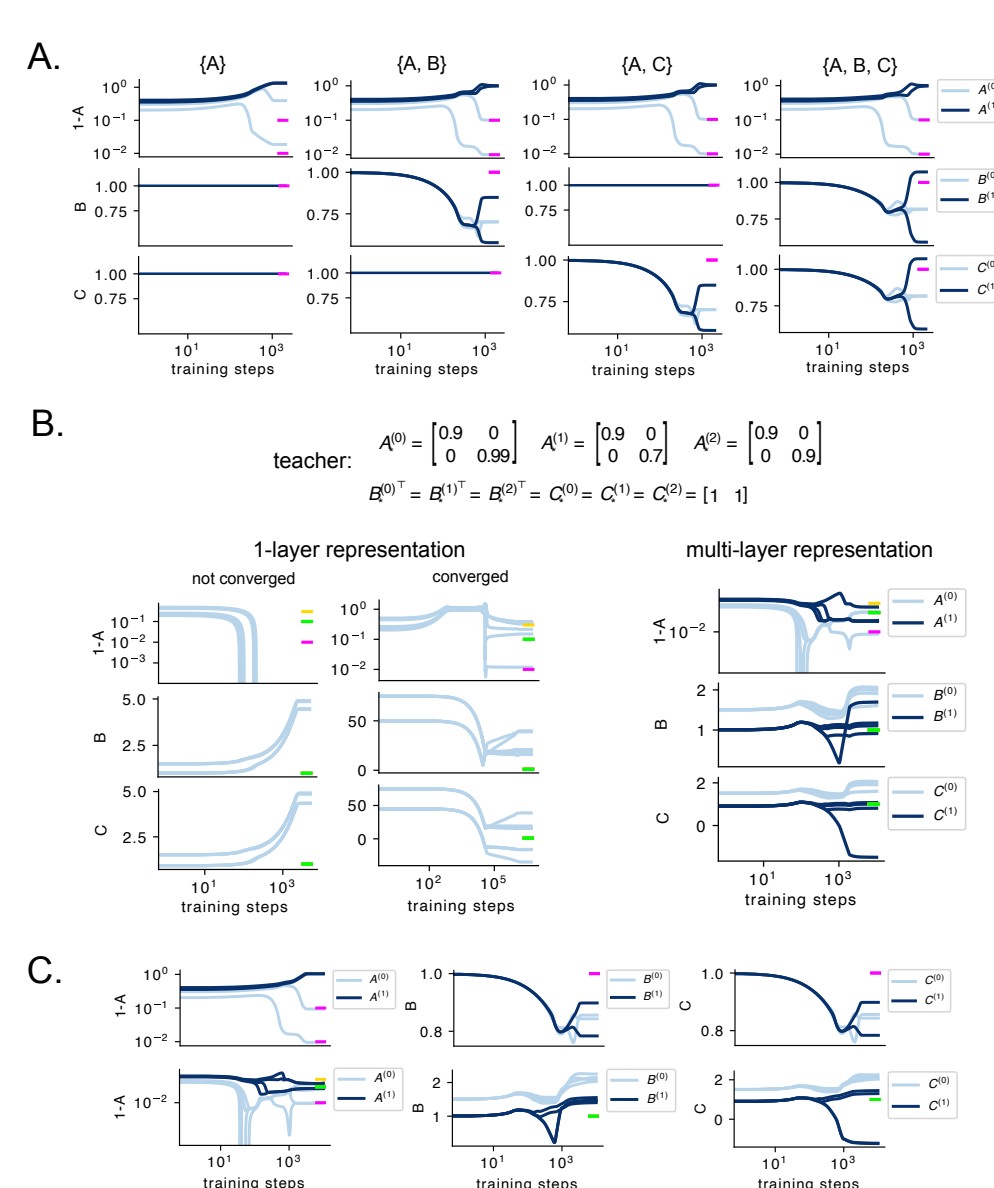

Figure 3: Teacher-student experiments. The inputs are uncorrelated white noise. Each student was trained with the Adam optimizer to match the teacher's outputs. The colored horizontal bars at the end of each subplot represent the teacher's parameters. Differently colored bars indicate different layers. **A.** Ablation experiments showing timescale learning across parameters of a deep SSM. A 2-layer student is trained to reproduce the outputs of a 1-layer teacher. Each column corresponds to an experiment where we train the parameter in $\{\cdot\}$ and keep the other parameters fixed. Each row illustrates how the student parameters evolve during learning. The teacher parameters are $A_* = \mathrm{diag}(0.9, 0.99)$ with $B = C^\top = \mathbf{1}_2$. **B.** Comparison between training an over-parameterized student model on a 3-layer teacher on two equivalent model representations derived in Section 3.3, the 1-layer vs multi-layer. Despite equivalent temporal expressivity, the 1-layer student is more sensitive to the initialization of $B$ and $C$, requiring large initial values to converge to a solution. In contrast, the 2-layer student learns within standard initialization range of $B$ and $C$ and consistently converges to a solution. The learning rate was initialized to $5 \times 10^{-5}$ in each case. **C.** Same experiments as in **A** and **B** but adding a ReLU activation after each layer of the student.

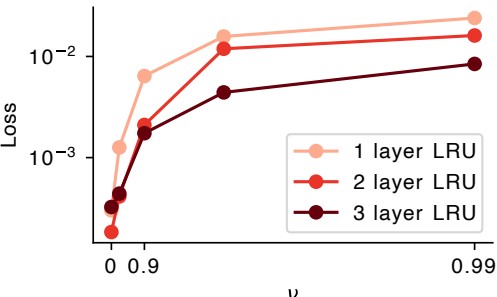

Figure 4: Depth improves the performance of the LRU trained on the teacher-student task with a fixed number of parameters. The horizontal axis shows the timescale parameter $\nu$ of the teacher (used as in (Orvieto et al., 2023)) to be learned by the student, each $\nu$ parameterized a new teacher dataset. Across sweeps of 1, 2, and 3 stacked LRU layers with a comparable number of parameters, the multi-layer representations outperformed the 1-layer model across all timescales of the teacher.

shown in the remaining panels. We observe the effects of $B$ and $C$ taken separately to be roughly equivalent in this setting, enabling the model to learn the exact eigenvalues of the teacher model at one layer without suffering the curse of memory, by regulating the latent signal in the feed-forward projections ($\{B_1^{(0)} = 0.70, B_2^{(0)} = 0.70, B_1^{(1)} = 0.57, B_2^{(1)} = 0.84\}$ for the experiment with $C$ fixed).

## 4.2 DEPTH VS WIDTH IN LEARNING CONVERGENCE

Section 3.3 discussed the equivalence between $K$-layer, $N$-dimensional state space models and a single layer, $KN$-dimensional system. When both $B$ and $C$ are learned, these two representations should be able to express the same temporal dependencies. Here we studied if this is true in practice using the experimental teacher-student paradigm.

We trained a student model with $K = 2$, $N = 4$, as well as a student with one layer and $N = 8$ dimensional states. The teacher was a $K = 3$-layer deep linear SSM with latent state size of $N = 2$. The two student models were overparameterized with respect to the teacher, rendering the task solvable. Figure 3B shows the exact parameterization of the teacher model and the results of the training runs. Both students were able to learn this task, as expected given their equivalent representational capacity.

However, the empirical results also suggested that two students learn differently. The left panel in Figure 3B shows two training runs of the single-layer student, a failed run and a successful run. The result appeared to be influenced by the initialization of the feed-forward projections $B$ and $C$. In the failed run, these parameters were initialized with values of $\mathcal{O}(1)$ for both students, following standard practice Gu et al. (2022c). In that regime, however, the eigenvalues of $A$ diverged to regions outside the edge of stability. By sufficiently amplifying the initialization of $B$ and $C$ (to $\mathcal{O}(50)$, found empirically), we found that the eigenvalues of $A$ remained stable throughout learning. Generally, we found that the single-layer student's learning convergence was more sensitive to the initial values of the feed-forward projections than the multi-layer parameterization.

## 4.3 TEACHER-STUDENT EXPERIMENTS WITH NONLINEARITIES

So far, we have focused on the learning dynamics of deep stacks of linear SSMs without position-wise nonlinear connections. To test whether these theoretical predictions generalize to deep nonlinear SSMs, we included a ReLU activation between each student layer. Figure 3C shows two examples replicating the experiments from earlier panels. Specifically, the first row of Figure 3C shows the effects of learning $B^{(k)}$ and $C^{(k)}$ on learning the particular timescales in the hidden state transition matrices $A^{(k)}$ across layers. Similarly, the addition of a nonlinearity did not affect the learning outcomes or convergence time of the student model from Figure 3B, shown in the second row of Figure 3C.

Finally, we tested whether these predictions generalize to models used in practice by training an LRU (Orvieto et al., 2023) on the teacher-student task. In this setting, we found that the benefits of depth over width in improving training convergence continued to apply, as shown in Figure 4. The details of the experimental setup are in Appendix C.1.

## 5 DISCUSSION

We analyzed the role of depth and width in learning long-term dependencies with deep state space models. By considering three measures of memory, we developed a new theoretical understanding of how deep SSMs encode long timescales and how parameterization in latent state size and the number of layers can affect learning. We showed that across the layers of a deep linear state space model, the autocorrelation function diverges as the eigenvalues of a latent dynamics matrix $A$ go to one, which in turn causes the variance of the latent states to diverge. These findings extend our understanding of the challenge of learning long-term dependencies with recurrent models, known as the curse of memory (Zucchet & Orvieto, 2024).

On the other hand, we found that *depth* might help to alleviate the curse of memory by allowing diagonal state space models to share timescales across layers. This allows deep SSMs to capture global long-term dependencies with small eigenvalues of the dynamics matrices $A^{(l)}$ at each layer $l$. This analysis used the notion of group delay as a proxy for memory in deep SSMs. Using the same theoretical framework, we analyzed the role of *width* in diagonal state space models in capturing temporal dependencies. This revealed the role of feedforward projections $B$ and $C$ in the group delay of the system and suggested how these parameters may regularize the eigenvalues of $A$ in practice. We validated our theoretical results in a synthetic teacher-student setting. Our empirical study of this task revealed regimes in which depth may reduce the model's sensitivity to parameter initialization and improve learning of long-range dependencies. This suggests there is still a theoretical gap to be filled in fully explaining the role of depth and width in learning convergence and initialization sensitivity. We plan to address these open questions in future work.

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

## A  PROOFS

**Lemma 1.** *Adapted from* [Zucchet & Orvieto](#) *(2024). For $\alpha, \beta \in \mathbb{C}$ satisfying $|\alpha| < 1$ and $|\beta| < 1$, and $(u_n)_{n \in \mathbb{Z}}$ a bounded sequence , we have*

$$\sum_{n,m \geq 0} \alpha^n \beta^n u_{n-m} = \frac{1}{1 - \alpha\beta} \left( u_0 + \sum_{\Delta \geq 1} (\alpha^\Delta u_\Delta + \beta^\Delta u_{-\Delta}) \right) \tag{12}$$

*Lemma 1.* We split the summation over indices $n$ and $m$ considering the cases where $n = m$, $n > m$, and $n < m$.

$$\sum_{n,m \geq 0} \alpha^n \beta^m u_{n-m} = \sum_{n=m} \alpha^n \beta^m u_0 + \sum_{n>m} \alpha^n \beta^m u_{n-m} + \sum_{n<m} \alpha^n \beta^m u_{m-n} \tag{13}$$

$$= \sum_n \alpha^n \beta^n u_0 + \sum_{m=0}^{\infty} \alpha^m \beta^m \sum_{\Delta \geq 1} \alpha^\Delta u_\Delta + \sum_{n=0}^{\infty} \alpha^n \beta^n \sum_{\Delta \geq 1} \beta^\Delta u_{-\Delta} \tag{14}$$

$$= \sum_n \alpha^n \beta^n \left( u_0 + \sum_{\Delta \geq 1} \alpha^\Delta u_\Delta + \sum_{\Delta \geq 1} \beta^\Delta u_{-\Delta} \right) \tag{15}$$

$$= \frac{1}{1 - \alpha\beta} \left( u_0 + \sum_{\Delta \geq 1} (\alpha^\Delta u_\Delta + \beta^\Delta u_{-\Delta}) \right) \tag{16}$$

$\square$

*Proposition 1.* Given

$$x_{t+1} = \lambda x_t + u_{t+1} \tag{17}$$

with $u_t$ satisfying wide-sense stationarity (WSS), we want to find a form for the auto-correlation function $R_x(\Delta) = \mathbb{E}[x_t x_{t+\Delta}]$. Due to the linearity of the system, the output is also WSS, which justifies considering $R_x(\Delta)$. We have that

$$x_{t+\Delta} = \sum_{n=0}^{\infty} \lambda^n u_{t+\Delta-n} \tag{18}$$

then

$$\mathbb{E}[x_{t+\Delta} x_t] = \mathbb{E}\left[ \left( \sum_{n=0}^{\infty} \lambda^n u_{t+\Delta-n} \right) \left( \sum_{m=0}^{\infty} \lambda^m u_{t-m} \right) \right] \tag{19}$$

$$= \sum_{n=0}^{\infty} \sum_{m=0}^{\infty} \lambda^n \lambda^m \mathbb{E}[u_{t+\Delta-n} u_{t-m}] \tag{20}$$

$$= \sum_{n=0}^{\infty} \sum_{m=0}^{\infty} \lambda^n \lambda^m R_u(\Delta - n + m) \tag{21}$$

Using Lemma 1 on the sequence $(R_u(\Delta - n))_{n \in \mathbb{Z}}$ with $\alpha = \beta = \lambda$, we obtain

$$R_x(\Delta) = \frac{1}{1 - \lambda^2} \left( R_u(\Delta) + \sum_{\Delta' \geq 1} \lambda^{\Delta'} (R_u(\Delta + \Delta') + R_u(\Delta - \Delta')) \right) \tag{22}$$

This gives the formula from the position, as $R_x = R^{(k)}$ and $R_u = R^{(k-1)}$. $\square$

*Proposition 2.* Unrolling $y_t$ in equation 1 for all points in the sequence with $x_0 = 0$, the SSM can be represented by a convolution, given by the impulse response of the linear time-invariant system,

$$y_t = (h * u)_t = \sum_{i=1}^{t} C A^{t-i} B u_i. \tag{23}$$

To efficiently compute the outputs of the convolution, Gu et al. (2022a) made use of the discrete Fourier transform to compute the outputs in the frequency domain before projecting back to the desired state-space. The mapping between the outputs in the time and frequency domains is afforded by the discrete convolution theorem. Let $Y_k = \mathcal{F}(y_t)$, $U_k = \mathcal{F}(u_t)$, where $\mathcal{F}$ is the discrete Fourier transform. Then

$$Y_k = \mathcal{F}\left(\sum_{i=1}^{t} C A^{t-i} B u_i\right) \tag{24}$$

$$= \sum_{t=-\infty}^{\infty} \left(\sum_{i=1}^{t} C A^{t-i} B u_i\right) e^{-j\frac{2\pi k}{L}t} \tag{25}$$

$$= \sum_{t=i}^{\infty} \sum_{i=1}^{\infty} C A^{t-i} B u_i e^{-j\frac{2\pi k}{L}t} \tag{26}$$

$$= \sum_{m=0}^{\infty} \sum_{i=1}^{\infty} C A^{m} B u_i e^{-j\frac{2\pi k}{L}(m+i)} \tag{27}$$

where we applied the transformation $m = t - i$ and noted that $u_t$ is defined for $t \geq 1$. Rearranging terms and using the definition of the DFT, $U_k = \sum_{i=1}^{\infty} u_i e^{-j\frac{2\pi k}{L}i}$, we obtain,

$$Y_k = \sum_{m=0}^{\infty} C A^{m} B e^{-j\frac{2\pi k}{L}m} \sum_{i=1}^{\infty} u_i e^{-j\frac{2\pi k}{L}i} \tag{28}$$

$$= C \left(\sum_{m=0}^{\infty} A^{m} e^{-j\frac{2\pi k}{L}m}\right) B U_k \tag{29}$$

Assuming $|A e^{-j\frac{2\pi k}{L}}| < 1$, which is a necessary condition for the stability of linear time-invariant systems, we can use the geometric series formula,

$$\sum_{m=0}^{\infty} \left(A e^{-j\frac{2\pi k}{L}}\right)^{m} = \left(I - A e^{-j\frac{2\pi k}{L}}\right)^{-1}. \tag{30}$$

Substituting this into the expression for $Y_k$,

$$Y_k = C(I - A e^{-j\frac{2\pi k}{L}})^{-1} B U_k = H_k U_k \tag{31}$$

where we have defined $H_k = C(I - A e^{-j\frac{2\pi k}{L}})^{-1} B$ as in Proposition 2. $\qquad\square$

## B  GROUP DELAY OF THE SYSTEM

Here we provide a detailed derivation of the group delay. Consider the frequency response of a scalar cascaded LTI system with $B = C = 1$,

$$H(z) = \prod_{l=1}^{K} \frac{1}{1 - \lambda^{(l)} z^{-1}} \tag{32}$$

Table 1: Comparison of depth and width on LRU performance in the teacher-student task. Each student was a model containing an LRU component with feedforward encoder and decoder. The students were trained using the Adam optimizer with learning rate $1e-3$.

| # Learnable Params | Learn $\{B, C\}$ | # of layers | Final Test Loss |
|---|---|---|---|
| 201 | No | 1 | 0.01006 |
| 196 | No | 2 | 0.00722 |
| 2374 | Yes | 1 | 0.00709 |
| 2597 | Yes | 1 | 0.00729 |
| 2446 | Yes | 2 | 0.00034 |

where $z = e^{j\omega_k}$. The group delay $\tau_d$ is defined as the negative of the derivative of the phase of the transfer function with respect to frequency, evaluated at $\omega_k = 0$.

$$\log H(z) = -\sum_{l=1}^{K} \log\left(1 - \lambda^{(l)} z^{-1}\right) \tag{33}$$

$$\varphi(\omega) = -\sum_{l=1}^{K} \text{Im}[(1 - \lambda^{(l)} e^{-j\omega})] \tag{34}$$

$$= -\sum_{l=1}^{K} \arg(1 - \lambda^{(l)} e^{-j\omega}) \tag{35}$$

$$= -\sum_{l=1}^{K} \arctan \frac{-\lambda^{(l)} \sin\omega}{1 - \lambda^{(l)} \cos\omega} \tag{36}$$

$$\tau_d = -\frac{d\varphi(\omega)}{d\omega}\big|_{\omega=0} = \sum_{l=1}^{K} \frac{\lambda^{(l)}}{1 - \lambda^{(l)}} \tag{37}$$

It follows that the group delay is related to the sum of the time constants of the subsystems. It is however important to note that we get this intuitive result by looking at the group delay and not the characteristic time constant of the system, which is equal to the largest recurrent parameter $\lambda$ of the system.

## C   FURTHER RESULTS FROM TEACHER-STUDENT TASK

### C.1   LRU EXPERIMENTAL DETAILS

We train stacks of Linear Recurrent Units (LRU, (Orvieto et al., 2023)) on the teacher-student task described in Section 4. We sample random teachers with temporal dependencies controlled by the $\nu$ parameter and we train on LRU students of varying depth and width while keeping the total number of learnable parameters within the same order of magnitude. Table 1 shows the results from one of these experiments, showing that a 2-layer LRU model outperforms the 1-layer models in learning to reproduce the teacher's outputs. A more comprehensive result is shown in Figure 4, where we show the mean loss obtained from 5 experiments with different random seeds across a sweep of timescales of the teacher. Unsurprisingly, all models gradually perform worse as $\nu \to 1$, however, the multi-layer models consistently outperform the 1-layer model, even though all three models are in the same parameter regime of roughly 2400 learnable parameters.

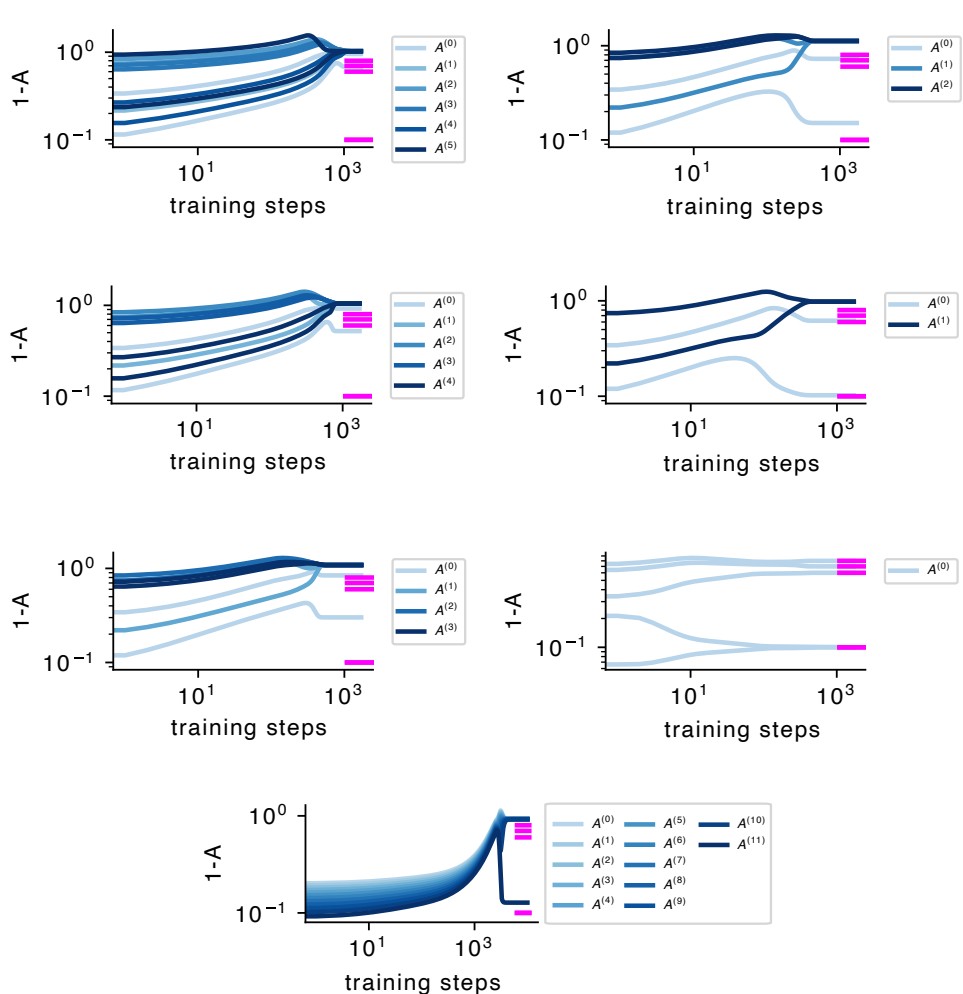

Figure 5: Adding depth allows a deep SSMs to capture long-term dependencies with smaller eigenvalues across layers. Successive experiments with the same teacher but different number of layers and latent state sizes for the student. The teacher's eigenvalues are $A = \mathrm{diag}(0.9, 0.4, 0.9, 0.3, 0.2)$ with $B = C = 1$. Colored bars at the right-end of each subplot show the target eigenvalues of the teacher.

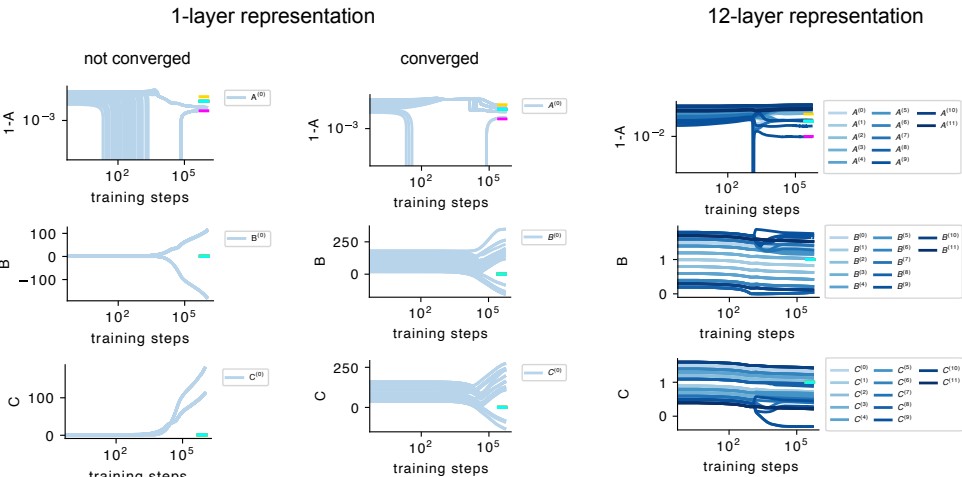

Figure 6: The teacher-student experiments shown in Figure 3 extended to deep SSMs with 12 layers. The task is defined by a 5-layer teacher SSM with a 2-dimensional latent state at each layer. The 12-layer, 1-dimensional student converges to a learned solution within $\mathcal{O}(10^5)$ training steps and all parameters remain in the vicinity of their initial values. To contrast, a 1-layer student SSM with the same number of parameters requires careful initialization of the $B$ and $C$ components to stabilize the eigenvalues of the dynamics matrix $A$ during training. The initial values for $B$ and $C$ were set to $\mathcal{O}(100)$ (middle panel). Initializing $B$ and $C$ within the standard initial value regime (left panel), $\mathcal{O}(1)$, results in most of the dynamics eigenvalues diverging outside the range of stability and the model does not converge in the given number of training steps.

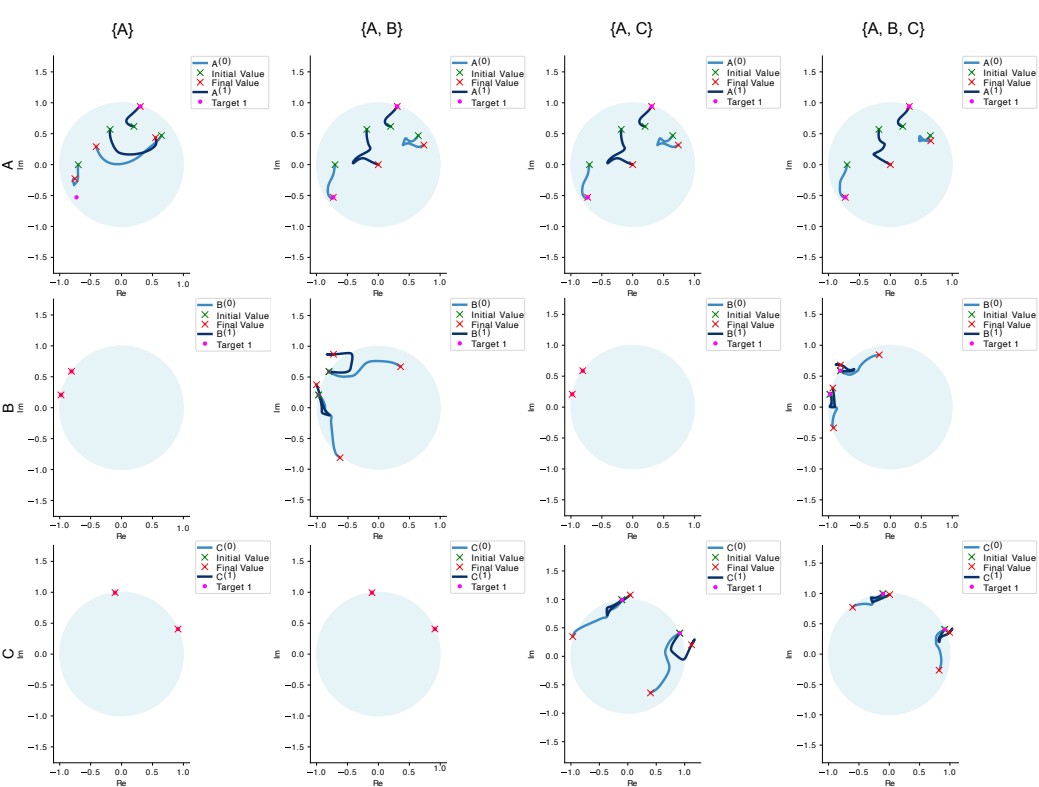

Figure 7: Teacher-student experiments with complex-valued state space models exhibit similar behaviors to their real-valued counterparts. The plots show ablation experiments similar to those shown in Figure 3A. The student model values were initialized in the complex plane within the unit circle. Trajectories show the evolution of parameters during training. Column headers indicate which parameters were trained.

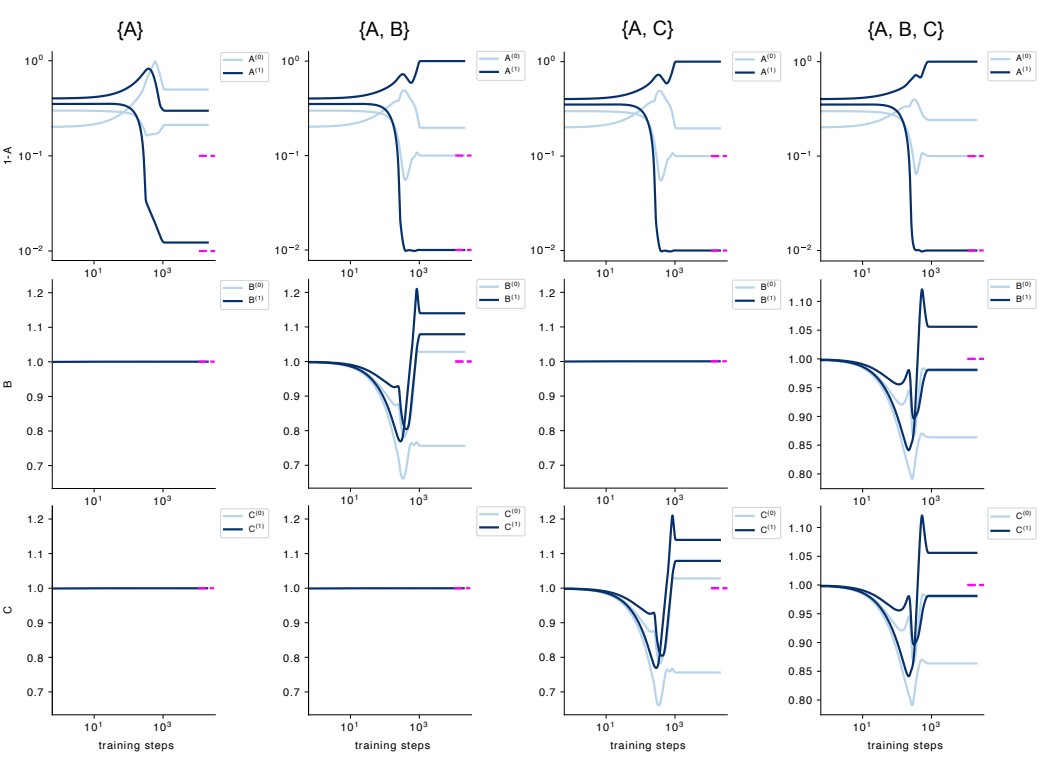

Figure 8: A projection of teacher-student experiments described in Figure 7 to real values shows the effects of learning $B$ and $C$ on eigenvalues of the $A$ matrices across layers. The projected real-valued trajectories are comparable to those observed in the real-valued SSM experiments from Figure 3.

