# OpenReview forum: "On the interplay between learning and memory in deep state space models"
_ICLR.cc/2025/Conference — Submitted to ICLR 2025_

### Official Review · Reviewer_m1Wn · 2024-10-17

**Soundness:** 2
**Presentation:** 2
**Contribution:** 1
**Rating:** 3
**Confidence:** 5

**Summary:**

This paper investigates memory in deep state-space models. However, the analysis is limited to linear state-space models, which presents a significant gap compared to models commonly used in practice. This opens up a potential research direction to explore whether the phenomena observed in this paper are specific to linear models or hold more generally for nonlinear cases.

**Strengths:**

This paper provides a detailed analysis of deep state-space models, using the theory of linear state-space models to demonstrate the performance of recurrent models. It includes a thorough review of related work, and the topic of identifying the advantages of state-space models in learning long-term memory is indeed important. **However, I would expect the paper to engage more deeply with prior research rather than focusing primarily on synthetic examples, which may not offer much inspiration for theoretical advancements or practical applications**.

**Weaknesses:**

1. **Lack of Reference to Relevant Literature**: The paper’s calculation of the autocorrelation function of linear state-space models should reference time series literature (e.g., ARMA models). It doesn’t add new insight and lacks generalization to nonlinear models, leaving questions on how the results extend.
2. **Limited Scale of Phenomenon Studied**: The scope of the paper is too narrow, focusing on a small-scale phenomenon, making comparisons with larger, well-established models (like S5 and Mamba) less meaningful.
3. **Insufficient Depth in Experiments (Figure 4)**: The experiments only use 1-3 layers of LRU, which is far from practical applications. More layers (up to 12) should be tested to understand the true behavior of the model as depth increases. Given the layers used in Figure 2, it is believed that extending the layers up to 12 is feasible within the available computational budget.
4. **Unclear Key Message for Different Audiences**: The paper doesn’t clearly separate the key takeaways for theoretical researchers and practitioners, which weakens its impact for both groups.

**Questions:**

1. It is difficult to discern the key contributions of this paper in relation to the existing literature, such as Recurrent Neural Networks: Vanishing and Exploding Gradients Are Not the End of the Story (https://arxiv.org/abs/2405.21064). There are a lot of similarity in the key message. It is important to clearly demonstrate how your work advances the recent literature, as the key contributions of this paper are currently hard to identify.
2. Another issue is the significant duplication of content in the appendix for Lemma 1 (from the arxiv paper Recurrent Neural Networks: Vanishing and Exploding Gradients Are Not the End of the Story). If the proof has been adapted from previous work, it may be more appropriate to cite the original source instead.
3. In appendix Figure 5, the current argument is hard to show adding depth allows a deep SSMs to capture long-term dependencies with smaller eigenvalues across layers. Maybe there should be a bit more discussion.

---

> ### Author Response · Authors · 2024-11-19
>
> Thank you for providing constructive feedback and for your praise of our detailed analysis and thorough review of related work. We are also glad you agree that identifying advantages of SSMs in learning long-term memory is important. We will now address your concerns about the novelty and scope of our work and our engagement with prior time series literature.
>
> One major argument concerns the applicability of our work to nonlinear models. In the paper, we begin by referencing prior work that has taken a similar approach to studying the linearized versions of deep learning architectures, which already exhibit non-trivial properties, and later connect them with their nonlinear counterparts. We think this approach is beneficial for fundamental theory work as we can reduce the phenomenon of study to its simplest form, in our case learning long-term dependencies with deep SSMs. We develop our theoretical framework for studying how these models learn long-term dependencies and then connect our findings with experiments. We study the linear system identification task where a teacher model is used to generate input-output pairs to train a student model. Figure 3A-B shows experiments with the linear models described by the theory, however, Figure 3C extends this setting to include models with nonlinear activations between layers, which shows qualitatively the same training dynamics and outcomes for the nonlinear models.
>
> **Re prior work comparison** The goal of our paper is to understand how deep SSMs learn long-range dependencies. While our work shares some common ground with Zucchet and Orvieto’s (2024) paper, our contributions extend significantly beyond their analysis. Specifically:
> - We extend their curse of memory analysis for a single layer by incorporating the evolution of autocorrelation of inputs across many layers of a deep SSM
> - We introduce novel analysis of learned temporal dependencies across layers of deep SSMs, moving beyond the single-layer case that restricts analysis to eigenvalues of the dynamics (A) matrix
> - We demonstrate that analyzing single-layer dynamics gives an incomplete picture for deep SSMs, as we show that group delay affects the long-range dependencies learned by deep SSMs. We show this delay is affected by both the depth of the model and the B and C matrices, explaining their role in learning long-range dependencies. To our knowledge, this has not been addressed by any prior work.
>
> **Re depth of models in experiments** We agree it is important to include experiments with deeper models to solidify our analysis. We first note that Figure 5 in the appendix of the original submission shows one such experiment in which we trained deep SSMs with up to 6 layers, showing the decomposition of timescales across layers. The experiments we have done so far indicate that no new behavior beyond what we observed for the 3 layer case in the main paper is observed for deeper models. We have now included 12-layer deep SSM experiments in **Figures 5 and 6** in the appendix.
>
> **Re engaging with prior work** You point out an existing connection between the analysis of input autocorrelation functions (ACF) and state space models. We recognize that in conventional time-series analysis, the ACF provides a useful summary of a linear model, which can aid in model selection, fitting, and interpretation (Shumway & Stoffer, “Time Series Analysis and its Applications”, Fourth Edition). Here, we use the ACF to study the properties of deep SSMs, namely, the connection between the ACF across layers and the evolution of the hidden state variance across layers of the deep SSM.
>
> **Re duplication of content** You raised  concerns about duplication of content from prior work, namely, the paper Zucchet and Orvieto (2024). Note that our Lemma 1 differs from that of Zucchet and Orvieto (2024) by the assumption of symmetry of the inputs $u_t$, which also changes its proof. We included the full derivation so that the proof of proposition 1 is fully contained in the main paper. That being said, we have now included a citation in Lemma 1 to clarify that it is a variation on the lemma introduced in Zucchet and Orvieto (2024).
>
> **Re figure 5** In the original submission, Figure 5 showed six separate deep SSMs trained on the same linear system identification task generated by a one-layer teacher SSM. We show the learning trajectory of models with varying numbers of layers, starting at six, ending at one, with each model converging to a solution. The deeper SSMs learn eigenvalues that are generally smaller than those of the teacher, with the exception being the one-layer student, which converges to the teacher parameters exactly.
>
> Thank you again for your feedback. We believe we have addressed the concerns raised in the review and hope you will reconsider your assessment. Please let us know if we can provide further clarifications.

---

> ### Author Response · Authors · 2024-11-26
>
> Dear reviewer m1Wn,
>
> Thank you again for taking the time to review our paper and provide detailed feedback. Please let us know if we can answer any more questions before the paper revision deadline on November 27th. The deadline for you to submit any further comments is December 2nd.

---

> > ### Comment · Reviewer_m1Wn · 2024-11-27
> >
> > 1. Thank you for your clarifications and the additional details. After reviewing them, I have decided to maintain my score.
> > 2. After reading the comments from other reviewers, I have a small question regarding LRU regularization. The term “regularization” does not appear explicitly in the LRU paper, so I wonder if the concept of input regularization could be clarified further.
> > 3. Regarding layer normalization and figure 1C, I am curious whether the assumption of exponential variance could be relaxed to linear growth (with respect to depth). If so, this might no longer pose a significant issue for deep SSMs.
> > 4. I believe this is an important research direction. However, the current results on deep linear SSMs feel somewhat preliminary. I look forward to seeing efforts to extend these results to multi-layer nonlinear models, which could potentially help scale up SSMs more effectively.

---

> > > ### Author Response · Authors · 2024-12-03
> > >
> > > Thank you for the response and additional questions.
> > >
> > > 2. By input regularization, we mean the $\gamma$ term that's applied to the inputs of LRU layers. In Orvieto et al., Resurrecting Recurrent Neural Networks for Long Sequences, (2023), the term input normalization was used instead.
> > > 3. We note that the exponential growth of hidden state variance across depth of the model was an observation that followed from our derivations, it was not an assumption. With layernorm, the hidden state variance would remain constant across layers. This is achieved by effectively removing signals from latent state dimensions with small timescales (eigenvalues of the dynamics (A) matrix close to zero).
> > > 4. Our current experiments include multi-layer nonlinear models, which exhibit exactly the same behaviors as their linear counterparts, as we mentioned in the paper and in our response to your original comment. We would also like to highlight that theoretical work often analyzes linearized versions of models, as nonlinearities can make the math analytically intractable. Nevertheless, linearity does not imply that the insights gained are not relevant or applicable to nonlinear models, as shown in both prior work (Saxe et al., 2014) and in our paper.
> > >
> > > Thank you again for taking the time to review our paper and provide feedback.

---

### Official Review · Reviewer_ZW4U · 2024-10-30

**Soundness:** 2
**Presentation:** 2
**Contribution:** 2
**Rating:** 3
**Confidence:** 4

**Summary:**

This paper studies deep linear SSMs, and investigates several aspects of model memory and how it depends on stacking layers. For example, it is demonstrated that the autocorrelation function of a input stationary sequence can be computed as it is passed through the layers in the simple case of 1D state spaces, where one can observe that the hidden state variance explode exponentially. Another characterisation of memory is given by the group delay, which is shown again to increase as the depth of the model increases. Lastly, in the contrary direction the authors show that a narrow deep linear SSM is equivalent to a shallow and wide SSM. Some numerical tests were performed which confirms the equivalence, while suggesting there is some difference in optimisation.

**Strengths:**

The problem studied is an interesting one, namely how does deep SSMs differ from shallow ones. This is also timely as while the theory of shallow SSMs are relatively well understood, the advantages of deep SSMs are less studied.

**Weaknesses:**

There are a number of weaknesses to this paper, mainly on the implication of the results presented. The current paper reads like a collection of observations and simplified analysis, but with very little cohesion and overall message. Some of these issues are outlined in "questions" below. I believe that the paper in the current from is too preliminary, and lacks a concrete message to both theoreticians and practitioners.

The paper also suffers from clarify issues, e.g.:
1. Group delay explanation is not very clear as it is only loosely defined in words. If you are using the standard definition of group and phase delay, it would be useful to formally introduce them.
2. Equation (5) contains typos - $k$ should be $l$ or vice versa
3. Sec 3.3: Eq (6) describes a general $K$-layer SSM but the statement before it says $K=2$. This is rather confusing. I do not think the authors meant to take $K=2$ here and after.

**Questions:**

1. Fig 1C and associated discussion: The variance explosion is because of the following: for a stationary process, if the filter is not normalised (e.g. by $\ell_1$ norm), then the output will scale depending on the filter. However, in practice this would rarely be a problem because most deep architectures (deep SSM, transformers) have a layer norm or similar layer to scale this problem away. Can the authors comment on the significance of this observation and why it is interesting in practical architectures with normalisation?
2. The authors show (and this is straightforward) that a 1-dimensional hidden state deep linear SSM can be represented by a shallow but wide one. The equivalence makes sense, but then the authors claim that deep SSM have certain properties that are different from shallow ones. These properties are not due to optimisation, but the actual architecture under certain choices of the trainable parameters (e.g. exploding variance). This appears contradictory, since they are equivalent, there is also a choice of the shallow SSM that reproduces the exact behavior, and vice versa. This need to be explained.
3. Results discussed in lines 309-321 on page 6: the authors say that the results of the learned eigenvalues are "consistent" with their theoretical predictions. Can you pin-point exactly which theoretical results are consistent with these and why? The only consistency that I can see is that shallow and deep linear SSMs are equivalent, but this is not surprising or interesting. Moreover, the training dynamics is random, and so should the initial condition. How representative of these results are of the general case over multiple repeats?
4. The implication of the form of $P$ and $P^{-1}$ is not very clear at the end of the section 3. In fact, it does not resolve why the autocorrelations explosion is not contradicted by the equivalence of the models shown in (8).
5. Most diagonal SSM models are complex valued, not real valued. How do the results on page 7 change when they are complex numbers instead?
6. What is the practical importance of the findings of the variance explosion, the group delay, and the equivalence result on training/designing SSMs, that we currently do not know?

---

> ### Author Response · Authors · 2024-11-19
>
> Thank you for taking the time to read our paper and provide a detailed review. We will now respond to the concern that the paper lacks cohesion between individual theoretical analyses.
>
> We are interested in providing a theoretical account of how deep SSMs learn long-range dependencies, which we develop in order of increasing complexity.
> - First, we examined how stacking multiple layers in a deep SSM affects hidden state variance. Larger hidden state variance leads to sharp loss landscapes which hinders learning. We show that deeper architectures intensify the variance explosion problem (Zucchet and Orvieto (2024)) due to increased cross-layer signal correlation.
> - This naturally leads to our analysis of group delay in Section 3.2 to characterize signal propagation through stacked LTI layers. This reveals previously unrecognized mechanisms of memory beyond individual layer timescales that enable deep SSMs to capture long-range dependencies. To our knowledge, this is a novel contribution to the deep SSM literature.
> - Section 3.3 refines our analysis by proving the equivalence between deep and shallow SSMs. This framework yields two key insights:
>     1. Timescales are shared across layers in deep SSMs, suggesting the falsifiable prediction that they can learn long-range dependencies without always requiring near-unity eigenvalues at each layer.
>     2.  The B and C matrices mediate cross-layer interactions. While diagonalizing the A matrix in the wide, shallow form decouples the time evolution of each state dimension, the input-output map remains unchanged—revealing that B and C matrices mediate these cross-layer interactions. Though they don't affect A's eigenvalues, they influence group delay—and thus, the system's memory characteristics.
>
> **Re shallow-deep equivalence in training** We showed two equivalent representations of a deep SSM encode the same temporal dependencies. This equivalence allows us to more readily study the specific differences in training between these two representations, specifically a single-layer SSM and a deep multi-layer SSM. Our experiments show that when learning long-range dependencies encoded by a teacher model, the multi-layer SSM has favorable training dynamics. This is to be expected and can be analyzed through the lens of prior learning dynamics literature (Saxe et al., 2014). In the paper, we explain this discrepancy through the sensitivity of parameter initialization. Specifically, in a deep SSM, B and C are initialized within the standard range (around 1) and then stay within this range across all layers throughout training. If we took this learned deep SSM and applied the transformations discussed in Section 3.3, we would see that the diagonal one-layer (wide) SSM representation will have B and C with significantly larger values, which may explain the difficulty of training the model in this (temporally) equivalent representation.
>
> **Re typos** Thank you for pointing out the typo in Equation (5), indeed k should be used throughout the equation. We have also corrected the K=2 statement that was present in a previous version of the paper. The current version includes a general analysis for any number of layers K.
>
> **Re Figure 1C and layernorm** Thank you for pointing out a possible source of confusion in Figure 1C. Indeed, introducing a layernorm would mitigate the signal explosion to some extent, although it’s important to note that layernorm is applied with respect to the latent state dimension, not with respect to time, which is why even with layernorm, deep SSM architectures use techniques like input regularization (LRU) and discretization (S4, S5) to regulate hidden state variance across layers.
>
> **Re consistency of theoretical predictions** The lines you referenced in the main paper were specifically related to the prediction that in the teacher-student experiment in Figure 3A, the two-layer SSM learns eigenvalues that are smaller at each layer compared to those of the one-layer teacher. This is consistent with the role of group delay in learning long-range dependencies for deep SSMs. While there are still some open questions, we have provided an initial theoretical framework through which to study the role of B and C in more depth, afforded by the equivalence between deep and shallow linear SSMs and the P matrix transformation from equation (9). The implication of this is that B and C can regulate the group delay across layers, effectively allowing the model to regulate the effects of exploding hidden state variance, as analyzed in Section 3.1.
>
> Our analysis has so far focused on real-valued SSMs, which have become common-place for working with language data. Nevertheless, we are happy to include analyses of complex-valued models in the appendix in our final submission.
>
> Thank you again for providing thorough and constructive feedback. We hope we have addressed your major concerns. Please let us know if we can provide additional details or clarifications.

---

> ### Author Response · Authors · 2024-11-26
>
> Dear reviewer ZW4U,
>
> Thank you again for taking the time to provide detailed feedback on our paper. We have now added additional figures showing experiments with complex-valued SSMs to the appendix. Please let us know if we can answer any more questions before the paper revision deadline on November 27. The deadline for you to submit any further comments is December 2nd.

---

> > ### Comment · Reviewer_ZW4U · 2024-11-28
> >
> > Thanks to the authors for the additional explanations. I do not have further queries. While the additional explanations are useful, the paper in the current form (even with these explanations added), in my opinion, is interesting but too preliminary, I believe there is potential for this paper to be much more impactful with further work put into it. Hence, I will maintain my current recommendation.
> >
> > Here are some suggestions to consider in future versions of this paper.
> >
> > 1. Quantitative (not just qualitative) predictions from the theory, and verified by experiments of varying complexity (e.g. quantifying the group delay due to B/C matrices, and actually show in a real SSM that this is the case).
> > 2. Improve overall cohesion of the message and resolve any apparent contradictions (e.g. the sense of which deep and shallow linear SSMs are equivalent, and in what sense they differ that does not contradict their equivalence).
> > 3. Present concrete algorithmic improvements that result from these new understandings, and demonstrating them on standard benchmarks such as the long range arena, or language datasets.

---

> > > ### Author Response · Authors · 2024-12-03
> > >
> > > Thank you for your response. We appreciate your feedback and detailed suggestions.

---

### Official Review · Reviewer_XgTv · 2024-11-01

**Soundness:** 3
**Presentation:** 3
**Contribution:** 3
**Rating:** 6
**Confidence:** 4

**Summary:**

This paper analyzes various sources of modeling memory in a Linear State Space Model. The authors take three different perspectives on how the width of the SSM layers and the depth of the model affect the learning capabilities of linear SSM under the simplified settings when B and C matrices are constant.

The first perspective is that of memory and autocorrelation function, where the authors show that the correlation between inputs further apart in the time increases as the number of layers increases, showing that depth is beneficial when it comes to modeling memory.

The second perspective is that of learning the group delay response, where the authors show that for an impulse input, the response gets delayed as the the depth increases, as long as the input is appropriately normalized between the layers.

Finally, the authors study the case of a single layer (multidimensional) SSM that approximates a multi-layer single dimension SSM. In this case the authors show that since the $A$ matrix can be diagonalized, the resulting eigenvectors when multipled by $B$ appropriately delay the input signal.

**Strengths:**

The paper is quite insightful and uses simple and effective setups/experiments to explain how SSMs can model long term dependencies. It is also easy to follow barring few places where some of the terms introduced are not explained properly (I have pointed them in the weakness/questions section.)

Few things that I believe are important theoretical observations:
- If any layer learns a long term dependency by setting $\lambda \approx 1$ then some other layer in the network needs to learn a $\lambda$ value close to $0$ in order to ensure that the state variance does not explode.
- The effects of the eigenvectors on the input B matrix when considering a linear SSM is very insightful!!

The experiments are well designed, and show the resulting and expected behavior of the models very well!

**Weaknesses:**

- Some of the key terms in the paper are not well defined in the pain section of the paper.
	- In section 3.1, the authors do not define what R^{(k)}(\Delta) is, and it particular what $\Delta$ is. Judging by the derivation in the appendix and the previous work by Zucchet and Orvieto (2024) I believe that it is $E[X_{t+\Delta}X_{\Delta}]$ and that this quantity is independent of $t$.
- The details in the experiment sections are a bit unclear.
	- First of all, what is the motivation of the student teacher setup in the experiments. Is it mostly to generate $u_t$ and $y_t$ pairs. Should we assume that the teacher is a (untrained) planted model (i.e., with fixed A, B and C) and the job is mostly to give the subsequent target for the $u_t$?
- What is the key motivation of the student teacher setup in the experiments. Is it mostly to generate $u_t$ and consequent output pairs to teach
- In the experiments, the depth of the teacher considered is (maximum 3), and the students are even shallower. It would be nice to see the analysis done on more deeper networks, especially since the effects of depth is key in the theoretical results.
- The authors show results on LRU, some insight into what the model is and how the dynamics are different from the linear dynamics considered in the paper would be really appreciated!

**Questions:**

- What is the exact task that they authors are trying to solve for in their experiments. Should I just think of it as next step prediction, where are the $y_t$ coming from. Is the teacher trained/planted, and if it is trained where is $y_t$ coming from?
- In the experiments the authors choose the input to be uncorrelated white noise. How would the results change if the inputs were correlated somehow?
- Can the authors elaborate on the setups for the experiments in Figures 1 and 2, the inputs are clear, but not sure what the targets are to elicit the behavior of the layers.
- some questions regarding proposition 2:
	- What is the point of introducing proposition 2?
	- Unless I am mistaken, it seems that in the proposition A is considered to be the same throughout different layers in the network (i.e, A is constant). However the final assumption and the derivation considered in the paper assumes that A^{(k)} changes with the layers. Am I missing something. The setup is a bit unclear, since in Section B of the appendix the authors assume what the form of H(z) is directly.
	- Would appreciate some clarification here!

---

> ### Author Response · Authors · 2024-11-19
>
> Thank you for the detailed and constructive feedback. You highlighted the contribution of this work to the understanding of how deep SSMs learn long-range dependencies. We are glad you found our analysis of how different components of deep SSMs contribute to learning long-range dependencies insightful. We will now address the individual comments and questions.
>
> **Re teacher-student experiments** The goal of the teacher-student experiments is to study how known dynamical systems are learned by deep SSMs of varying latent state size and layer depth. The task is to map white noise inputs $u_t$ to the teacher-generated targets, $y_t$. The teacher is a fixed SSM with parameters chosen such that it encodes a dynamical system with long-term dependencies, i.e., with eigenvalues λ≈1. Importantly, this experimental setting allowed us to show that a deep SSM student can learn smaller timescales at each layer while still capturing long-term dependencies, thanks to the effects of group delay. This shows that understanding deep SSMs requires considering both group delay and layerwise dynamics.
>
> **Re model depth in experiments** Thank you for the suggestion on increasing the depth of the models used in our experiments. We have now extended our experiments to 12 layers in the appendix, showing similar effects of depth (and group delay) on learning long-term dependencies with deep SSMs.
>
> **Re Section 3.1** Yes, your definition of R^{(k)}(\Delta) is what we used in the paper. We will make sure to clarify this definition in the final version.
>
> **Re Figure 1 and 2** The role of Figures 1 and 2 is to study the general properties of signal propagation in deep SSMs, supplementing the theoretical analysis in Section 3. The models used to produce these figures were not trained on any particular task, their parameters were fixed manually. The model outputs in Figure 2A-B were produced by passing an impulse input (one-hot vector) and a white noise input through the fixed model in Figure 2C.
> - Figure 1 grounds our understanding of the curse of memory problem for deep SSMs, extending the analysis from (Zucchet and Orvieto, 2024). Specifically, we show that the inputs to layers in a deep SSM become more correlated regardless of the initial input autocorrelation function. To connect this to our experiments with white noise inputs, this shows that for sufficiently deep SSMs, changing the correlation of the inputs would not have a significant effect on the nature of the experimental setup. We will include this in our final submission.
> - Figure 2 then shows the effect of stacking multiple state space layers on the propagation of the signal for a simple impulse and white noise, both showing the effective delay that's introduced, complementing our theoretical analysis of group delay.
>
> **Re LRU experiments** The final experiment in our paper trains varying parameterizations of the Linear Recurrent Unit (LRU) model on the teacher-student task, adapted from Zucchet and Orvieto (2024). The LRU is a deep SSM architecture with input regularization designed to ameliorate the explosion of hidden state variance across the network. We trained multiple LRU models, varying the number of layers and their latent state size while keeping the number of trainable parameters roughly equal. The goal of this experiment was to reproduce the setting described in Figure 3B while using a deep SSM architecture used in practice. This experiment reproduces the same result shown in Figure 3B for the LRU, highlighting the differences between training parameter-equivalent representations of a deep SSM. We will include these details in our final submission.
>
> **Re proposition 2** Proposition 2 describes the form of a deep SSM in the frequency domain, specifically through the frequency response. The frequency response is later used to derive the form of the group delay, upon which we build our theoretical analysis of how deep SSMs learn long-range dependencies. We recognize that the relevance of proposition 2 might not be obvious without considering the derivation of group delay found in the appendix. We will make this connection more clear in the final version. We also thank the reviewer for requesting more clarity around this section. The parameters can generally vary across layers, in which case we would use the ^{(k)} superscript to distinguish parameters at layer k.
>
> Thank you for taking the time to review our paper. We hope this response addresses your questions. Please let us know if you have remaining questions we can clarify.

---

> > ### Comment · Reviewer_XgTv · 2024-11-28
> > **Response to the Rebuttal**
> >
> > I thank the authors for their rebuttals and answering my questions! I have no further questions to ask. I will keep my current score. Thanks!

---

> > > ### Author Response · Authors · 2024-12-03
> > >
> > > Thank you again for taking the time to review our paper and provide detailed feedback!

---

> ### Author Response · Authors · 2024-11-26
>
> Dear reviewer XgTv,
>
> Thank you again for taking the time to provide detailed and constructive feedback. Please let us know if we can answer any more questions before the paper revision deadline on November 27th. The deadline for you to submit any further comments is December 2nd.

---

### Author Response · Authors · 2024-11-19

# General response
We thank all three reviewers for taking the time to read our paper and provide thorough feedback. We are grateful for the depth of the reviewer’s comments and constructive criticism. We summarize the common themes across all reviews here and then provide individual responses to each reviewer. The reviewers generally recognize the importance of our work in understanding how deep state space models (SSMs) learn long-range dependencies, acknowledging that prior work was mostly restricted to studying the properties of single-layer models which omit the effects of depth. Furthermore, reviewer XgTv praised our methodical approach. They specifically liked our focus on simple experimental setups that complement our theoretical derivations while effectively addressing the role of different deep SSM parameters on learning long-range dependencies. Reviewer ZW4U raised a concern about the connections between subsections of our theoretical analysis and our experimental validation, a point partially echoed by reviewer m1Wn in their comment questioning the relevance of our work to different audiences. We address some of these concerns here and go into more detail in individual responses.

**Contributions and novelty**

There is growing empirical evidence suggesting that deep SSMs are well-suited to model tasks with long-range dependencies. Understanding how this is achieved is important for both the machine learning community to inform the development of future architectures as well as for scientists using these models in their research. The linear dynamics represented by individual layers of deep SSMs are appealing for their inherent interpretability as dynamical systems, however, theoretical understanding of deep stacks of these layers is still lacking.

**Theoretical Results**

We provide a brief summary of our theoretical results and contributions.

Our work aims to explain how deep SSMs are able to learn long-range dependencies. We begin by analyzing the explosion of hidden state variance for a single-layer SSM encoding long-term dependencies, a problem known as the curse of memory (Zucchet and Orvieto, 2024). We demonstrate that adding layers increases signal correlation, exacerbating hidden state variance explosion - a finding that may seemingly hinder the learning of long-term dependencies with deep SSMs.

Through an examination of group delay, we identify additional factors beyond layer-specific timescales that contribute to learning long-range dependencies in deep SSMs. Motivated by this, we leverage the equivalence between K-layer, 1-dimensional SSMs and 1-layer, K-dimensional SSMs, which enables a unified analysis that highlights the role of the input (B) and pass-through (C) matrices. This equivalence reveals that timescales are effectively shared across layers and suggests deep SSMs can learn long-range dependencies without requiring near-unity eigenvalues of the dynamics matrices (A) at each layer. Furthermore, diagonalization of the A matrix in the shallow, wide representation decouples latent state evolution, showing that the B and C matrices mediate cross-layer timescale interactions without affecting the system's eigenvalues, instead influencing the group delay. In conclusion, we show that temporal dependencies learned by deep SSMs are influenced by more than just the individual linear dynamical systems at each layer.

To our knowledge, our work is the first to address the role of multiple layers in deep SSMs in learning long-range dependencies by analyzing both explicit timescales encoded by the linear dynamics at each layer and the implicit delay that arises when stacking linear time-invariant systems. This answers an important question about the role of B and C matrices in deep SSMs, which we show modulate the implicit delay that arises from stacking multiple linear layers in a deep SSM.

---

### Meta-Review · Area_Chair_fP8s · 2024-12-22

**Metareview:**

This work addresses learning long-range dependencies in deep state-space models (SSMs), analyzing variance explosion and group delay in linear systems. While the work provides insights into theoretical aspects, reviewers identified several shortcomings. The results are preliminary, focused on linear systems with limited extension to practical, nonlinear cases. Experiments lack sufficient technical depth, the connection between theoretical and empirical results is a bit unclear, and broader implications remain underdeveloped. The rebuttal improved clarity on some issues but did not convincingly resolve concerns about the scope and impact. Given these limitations, I recommend rejection. That said, I believe the paper will be a stronger contribution after necessary revisions.

**Additional Comments On Reviewer Discussion:**

During the rebuttal, reviewers highlighted gaps in cohesion, experiment depth, and connections to nonlinear cases. Authors addressed some concerns, adding experiments with deeper models and extending analyses. However, responses did not fully resolve core issues like practical implications or extending findings to nonlinearity. Reviewers m1Wn and ZW4U maintained their stance, emphasizing the paper’s preliminary nature and the need for further development. Based on these, the conclusion is that the current submission needs more work before acceptance.

---

### Decision · Program_Chairs · 2025-01-22

Reject